# Striping of orbital-order with charge-disorder in optimally doped manganites

Wei-Tin Chen [1,2✉], Chin-Wei Wang[3], Ching-Chia Cheng[1], Yu-Chun Chuang [3], Arkadiy Simonov [4], Nicholas C. Bristowe [5] & Mark S. Senn [6✉]

The phase diagrams of $LaMnO_3$ perovskites have been intensely studied due to the colossal magnetoresistance (CMR) exhibited by compositions around the $\frac{3}{8}^{th}$ doping level. However, phase segregation between ferromagnetic (FM) metallic and antiferromagnetic (AFM) insulating states, which itself is believed to be responsible for the colossal change in resistance under applied magnetic field, has prevented an atomistic-level understanding of the orbital ordered (OO) state at this doping level. Here, through the detailed crystallographic analysis of the phase diagram of a prototype system ($AMn_3^{A'}Mn_4^BO_{12}$), we show that the superposition of two distinct lattice modes gives rise to a striping of OO Jahn-Teller active $Mn^{3+}$ and charge disordered (CD) $Mn^{3.5+}$ layers in a 1:3 ratio. This superposition only gives a cancellation of the Jahn-Teller-like displacements at the critical doping level. This striping of CD $Mn^{3.5+}$ with $Mn^{3+}$ provides a natural mechanism though which long range OO can melt, giving way to a conducting state.

[1] Center for Condensed Matter Sciences and Center of Atomic Initiative for New Materials, National Taiwan University, Taipei 10617, Taiwan. [2] Taiwan Consortium of Emergent Crystalline Materials, Ministry of Science and Technology, Taipei 10622, Taiwan. [3] National Synchrotron Radiation Research Center, Hsinchu 30076, Taiwan. [4] Materials Department, ETH Zürich, Vladimir-Prelog-Weg 1-5/10, 8093 Zürich, Switzerland. [5] Centre for Materials Physics, Durham University, South Road, Durham DH1 3LE, UK. [6] Department of Chemistry, University of Warwick, Gibbet Hill, Coventry CV4 7AL, UK. ✉email: weitinchen@ntu.edu.tw; m.senn@warwick.ac.uk

The archetypal colossal magnetoresistance (CMR) systems are the manganite perovskites, however, phase diagrams such as $La_{1-x}Ca_xMnO_3$ (LCMO) remain controversial. A key regime has been identified around $x = \frac{3}{8}$ as it corresponds to a maximum in the MR effect[1]. For narrow-bandwidth manganites (e.g. $Pr_{1-x}Ca_xMnO_3$ and $La_{1-x}Ca_xMnO_3$) this effect is coincident with the observation of substantial phase segregation between, what are believed to be, metallic ferromagnetic (FM) phases, and insulating charge and orbital-ordered (OO) phases[2]. Neutron-diffraction studies of the magnetic-field-induced insulator-to-metal transition in $Pr_{0.7}Ca_{0.3}MnO_3$[3] and melting of charge ordering under irradiation[4] show that the delicate balance in these systems can be easily upset, providing an explanation for the colossal change in resistivity. However, the apparently intrinsic phase segregation in this regime of the OO phase diagram[5], make it hard to ascertain the microscopic structure of the insulating OO phase around $x = \frac{3}{8}$, which has been assumed to be essentially similar to that at $x = \frac{1}{2}$ (e.g. see models proposed for $x = 0.3$ [4,6]), consisting of a chequerboard $Mn^{3+}:Mn^{4+}$ charge ordering and CE-type orbital order[7]. However, this basic pattern of ordering is inconsistent with average valence state ($Mn^{3.375+}$) at this doping level, and as macroscopic charge segregation between the competing phases would be highly energetically unfavourable, a conundrum exists regarding where the missing charge resides.

A further key issue that has complicated the study of the phase diagram of the manganites is that the control parameter $x$ does not simply act to change the hole ($Mn^{4+}$) concentration, but also leads to band narrowing due significant changes in $MnO_6$ octahedral rotation and tilt angles caused by both varying ionic radii of dopants (e.g. $Ca^{2+}$ and $La^{3+}$) and that of $Mn^{3+}$ and $Mn^{4+}$ [8]. Here we show that the 134 perovskites $AMn_3^{A'}Mn_4^BO_{12}$, A = $Na_{1-x}Ca_x$ and $La_{1-x}Ca_x$ (with representative crystal structure shown in Fig. 1a), corresponding to the $x = 0 - \frac{1}{2}$ doped regime of the LCMO manganites, may be used as a prototype system to

revisit this long-standing problem. Our phase diagram, in which any significant band narrowing or intrinsic phase segregation is absent across the compositional range of interest, displays four orbital ordering regimes, including an as yet unobserved state consisting of striping of OO with charge disordered (CD) at $x \approx \frac{3}{8}$ corresponding with the optimal doping level in the CMR manganites.

## Results and discussion

We synthesise our prototype systems $AMn_3^{A'}Mn_4^BO_{12}$, A = $Na_{1-x}Ca_x$ and $La_{1-x}Ca_x$ at intervals $\Delta x = 0.1$ using high-pressure synthesis techniques (see section "Methods", Supplementary Table 1 and Supplementary Fig. 1), with A = La, Ca, Na corresponding to formal $Mn^B$ valence state of 3+, 3.25+ and 3.5+, respectively. We validate this formal assignment via bond valence sum (BVS)[9] analysis in Fig. 1c showing that doping on the A site has predominantly the effect of charge transfer to the B site, with the A' site maintaining a more-or-less constant valence. While at first glance introducing the additional transition metal on the A' site with respect to the $ABO_3$ perovskites may seem an undesirable complication, the increased chemical complexity results in an enhancement in physical structural attributes with respect to more conventional manganites, making them ideal for the present study. Firstly, the small changes in volume (<2%) and octahedral tilt angles (138.7° ± 1.2°) across the series (Fig. 1c) allows us to effectively decouple the band narrowing physics from the effect of electronic doping. Furthermore, since 3 in 4 of the A sites have well-defined long-range crystallographic order, we effectively reduce the average variance of the $AA'_3$ by a factor of 4 with respect to the A-site of the equivalent perovskite where this quantity has been shown to be an important control parameter[10]. Thirdly, the octahedral tilt pattern ($a^+a^+a^+$ in Glazer's notation[11]) in our prototype 134 system is locked in place by the A' site cation order[12] producing cubic lattice symmetry, which is in stark contrast to the orthorhombic $Pnma$ manganites ($a^-a^-c^+$). This fact will become particularly crucial in

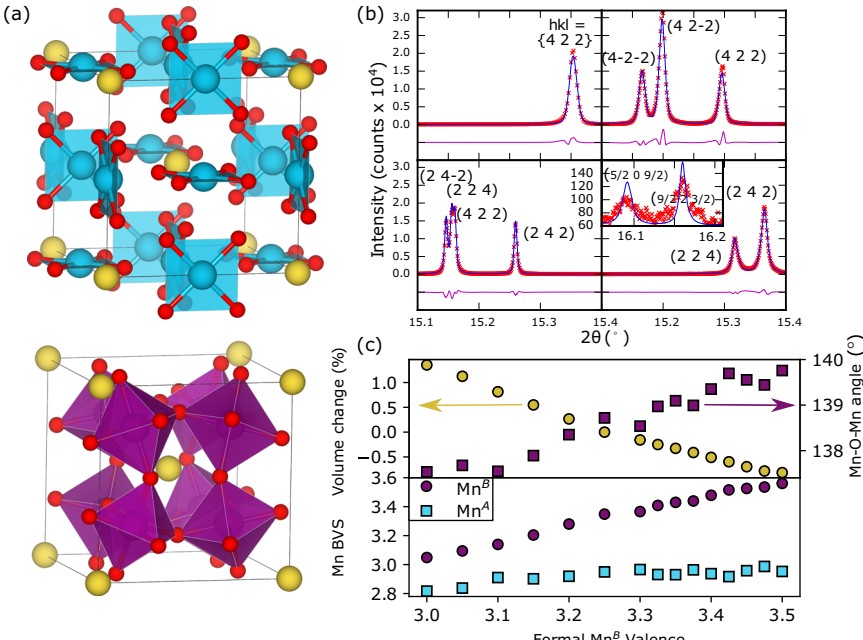

**Fig. 1 Structure an doping of the prototype system. a** The high-symmetry structure of the $AMn_7O_{12}$ prototype system with square planar coordinate $Mn^{A'}$ (top), and octahedrally coordinated $Mn^B$ (bottom) illustrated. The variable A-sites = $Na_{1-x}Ca_x$ and $La_{1-x}Ca_x$ used to dope the structure are shown as yellow spheres. **b** Diffraction data on $AMn_7O_{12}$, showing cubic $Im\bar{3}$ $Na_{0.9}Ca_{0.1}Mn_7O_{12}$ (top left, at 700 K), rhombohedral $R\bar{3}$ $La_{0.4}Ca_{0.6}Mn_7O_{12}$ (top right, at 80 K), monoclinic $I2/m$ $LaMn_7O_{12}$ (bottom left, at 80 K) and pseudo-tetragonal ($C2/m$) $Na_{0.4}Ca_{0.6}Mn_7O_{12}$ (bottom right, at 80 K) with inset superstructure peaks. Peaks are indexed with respect to $Im\bar{3}$ parent symmetry. **c** Change in volume, $Mn^B$–O–$Mn^B$ bond angle and bond valence sums (BVS) as a function of formal $Mn^B$ valence state at 700 K.

the discussion that follows since any further lattice distortion in our prototype system will be indicative of long-range orbital ordering. For example, Fig. 1b shows the cubic 134 aristotype symmetry evident by an unsplit profile of the (4 2 2) powder diffraction peak. On cooling, either a rhombohedral phase or one of two distinct kinds of predominately pseudo-tetragonal lattice distortions are observed (Fig. 1b), depending on the doping level. In each case, the underlying lattice distortions are coupled to the long-range orbital order which are themselves described by frozen-out normal modes of the lattice as discussed in detail later. Finally, and in part due to the aforementioned attributes, we find our 134 perovskite prototype systems to have a very high degree of crystallinity compared to simple perovskites of similar composition, with key compositions having average $\frac{100 \times \delta d}{d} = e_0 = 0.007$–$0.012\%$ (see Supplementary Table 1). This fact has enabled us to conduct a much more detailed and finely resolved structural study of the manganite phase diagram than has previously been possible, allowing a solution for this long-standing problem.

Four distinct crystallographic phases as a function of temperature and composition are evident from our variable temperature diffraction studies (Fig. 2 and Supplementary Fig. 2). Full details of the structural refinements are given in the "Methods" sections and in Supplementary Tables 1 and 2. At high temperatures (700 K), the aforementioned cubic aristotype is observed across all compositions. On cooling, the La-rich ($Mn_B^{3+}$) phase can be described as C-type orbital order with planes perpendicular to [0 1 0] (Fig. 2d). The JT distortions have the orthorhombic mode of the 2 short: 2 medium: 2 long Mn–O bond lengths, where the short and long bonds alternate along the $a$ and $c$ pseudo cubic lattice directions, and the medium is along $b$[13,14]. This phase is completely analogous to the OO phase of simple perovskite $LaMnO_3$, and the orbital order may be described as transforming as the irreducible representation $M_3^+$ of the parent $Pm\bar{3}m$ space group. A = Ca ($Mn_B^{3.25+}$), in which OO has previously been reported[15] in a rhombohedral structure, may also be viewed as C-type orbital (and charge order) but now with the planes of OO perpendicular to [1 1 1], preserving a 3-fold axis of the cubic aristotype. It contains three JT active $Mn^{3+}$ sites, which have 4 long: 2 short bonds, for every non-JT active $Mn^{4+}$ site (Fig. 2d). The apparent 4-long 2-short distortion is actually due to an averaging of disordered JT 2-long 4-short bonds about the [111] axis[16] that resolves itself in long-range incommensurate order at 250 K in A = Ca[17,18], which we find is washed out rapidly with doping (Fig. 2c). This rhombohedral phase persists up to a doping levels of $Mn_B^{3.325+}$ (A = $Ca_{0.7}Na_{0.3}$) beyond which point a pronounced change in lattice symmetry occurs. It is this region of the phase diagram, especially around $\frac{3}{8}^{th}$ doping level ($Mn_B^{3.375+}$) that is the focus of this paper since it corresponds with the maximum in the CMR of the much studied LCMO systems. At the Na-rich end ($Mn_B^{3.5+}$) additional superstructure reflections can be indexed on a propagation $\mathbf{k} = (\frac{1}{4} 0 \frac{1}{4})$ and may be fit well by atomic displacements transforming as irrep. $\Sigma_2$ (see Supplementary Figs. 3–5). This produces stripes of orbitally ordered $Mn^{3+}$ separated by $Mn^{4+}$ perpendicular to [1 0 -1] pseudo cubic axes. For the $Mn^{3+}$ stripes, the JT axes (i.e. the long Mn–O bond lengths) alternate along the crystallographic $a$ or $c$ directions. This results in a pattern that repeats itself every 4th stripe. Again, this model is consistent with the charge and OO phase of the famous half-doped manganites, and also a more recent crystallographic study of the 134 system[19]. However, our precise refinement of the superstructure peaks related to the order parameters that describes the orbital order across the phase diagram (see inset Fig. 1b and Supplementary Fig. 4) show that the amplitude of $\Sigma_2$ component shrinks while $M_3^+$ (C-type) grows in (Fig. 3), such that they approach equality at the $\frac{3}{8}$ doping level. The microscopic manifestation of the evolution and interference between these two lattice modes is such that only 1 in every 4 stripe displays a pronounced JT elongation, and now always along the pseudo cubic c-axis (see Fig. 2). This new pattern of JT distortions implies a melting of orbital ordering as the electron concentration is increased with respect to the $\frac{1}{2}$-doped phase, and is one of the most striking features of this phase diagram, implying either that orbital order ($Mn^{3+}$) and charge disorder ($Mn^{3.5+}$) coexist in neighbouring stripes, or indeed that there are striped regions of the structure that are locally itinerant in nature and can coexist on this microscopic length scale with those exhibiting orbital order and hence insulating behaviour. Our detailed single-crystal diffraction study at this doping level shows the absence of any significant diffuse scattering (Supplementary Fig. 6), pointing towards the latter, itinerant, scenario. Our observation of this feature at the precise doping level at which a maximum in the CMR response of the LCMO systems is observed provides mechanistic insight for how a bulk conducting phase might emerge from the insulating state via the formation of extended CD domain boundary walls (e.g. a single domain boundary wall could producing a CD region of ≈2.4 nm, twice the crystallographic modulation length). Understanding the domain structure, which is invariably rich in phases when multi-dimensional order parameters are involved (e.g. as in the hybrid improper ferroelectrics[20]), will no doubt provide further insight into how these stripes can coalesce to produce bulk conductivity.

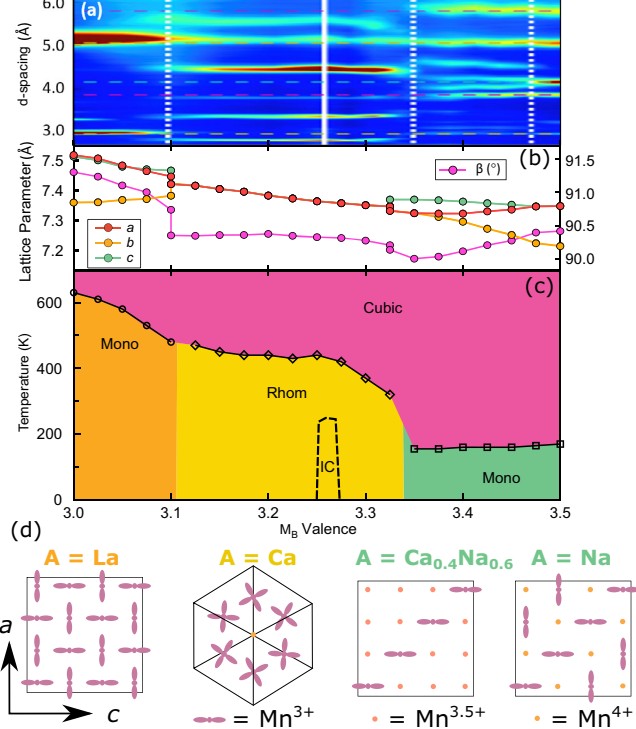

**Fig. 2 The experimental phase diagram of the $AMn_7O_{12}$ series, plotted as a function of formal B-site valence, constructed from 21 samples measured by synchrotron X-ray powder diffraction (SXRD) and neutron powder diffraction (NPD). a** Four distinct regions of the phase diagram are evident from the low-temperature low-$d$-spacing NPD data, plotted as a heat map. **b** Lattice parameters at 80 K evidence 4 kinds of metric symmetry. **c** the temperature-doping phase diagram constructed from the variable temperature SXRD and NPD data. IC denotes an incommensurate phase that is found for limited stoichiometric variation around A = Ca. **d** The orbital-ordered structures in the four distinct regimes determined in the present study.

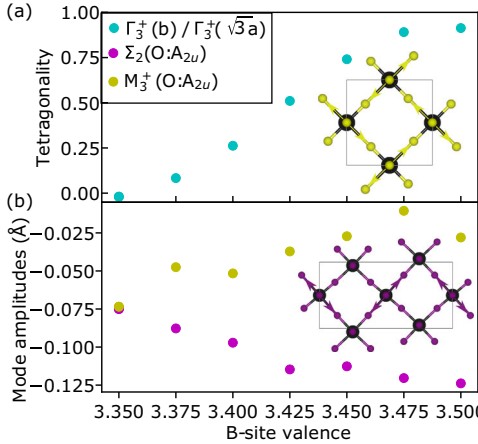

**Fig. 3 Parameters extracted from Rietveld refinement against high-resolution synchrotron X-ray powder diffraction (SXRD) data on $AMn_7O_{12}$. a** Symmetrised strain ($\Gamma_3^+$(b) / $\Gamma_3^+$($\sqrt{3}a$)) indicating that pseudo metric tetragonal symmetry is observed for a B-site valence around $\frac{1}{2}$ and $\frac{3}{8}$ doping levels. **b** The orbital ordering distortion modes transforming as irrep. $M_3^+$ and $\Sigma_2$ of $Pm\bar{3}m$ with Jahn-Teller distortion characters as shown in the insets. Large uncertainties are associated with determining $M_3^+$ since in the refinements it is highly correlated with octahedral tilt displacements ($M_2^+$). However, the valence-dependent trend and the strong correlations with the precisely determined strains that couple directly to $M_3^+$ and $\Sigma_2$ (see Supplementary Fig. 7), give a high degree of confidence in the results.

This scenario of continuously melting orbital order across the $x = \frac{1}{2}$ to $\frac{3}{8}$ doping level is highly consistent with the evolution of the macrostrain (change in lattice parameters) that we observe across the phase diagram. In order to analyse the coupling between OO and lattice strain more fully, we define a tetragonality parameter, $t$, which varies continuously from 1 to 0 for pure $b$-unique to pure $c$-unique tetragonal strain (see "Methods"). This plot of $t$ (see Fig. 3a) clearly reveals two regimes of pseudo symmetry in this range of our phase diagram. Firstly, at the Na-rich end $t = 1$, the coupling of orbital order to strain produces a pseudo-tetragonal state (see Fig. 3), which is the same as at the La-rich end (Supplementary Figs. 7 and 8), and arises as an equal number of JT long-axes ($d_{3z^2-r^2}$ or $d_{3x^2-r^2}$) point along $a$ and $c$, with no elongations occurring along $b$. This leads to a pattern of 2-long, 1-short of the crystallographic axes (see Fig. 1). However, towards $A = Na_{0.4}Ca_{0.6}$, there are now only JT distortions along the $c$-axis, leading to a 1-long 2-short pattern of the crystallographic axes. The continuous evolution of the macrostrain of the prototype system from $x = \frac{1}{2}$ ($A = Na$) to $\frac{3}{8}$ supports a gradual melting of the long-range JT order in half of the stripes, consistent with the mode amplitudes of $\Sigma_2$ and $M_3^+$ extracted from our Rietveld refinement against the diffraction (Fig. 3b). It is important to emphasise here that the microstrain in the sample remains exceptionally low across these phase transitions ($e_0 \leq 0.015\%$), precluding the existence of any significant intrinsic phase segregation. Additionally, since there are no tetragonal subgroups of the $Im\bar{3}$ aristotype (it lacking any 4-fold symmetry elements), the pseudo lattice symmetry reported here (to within a macrostrain of 0.02%, see Supplementary Fig. 9) must be taken as a hallmark of the underlying configuration of the OO. Indeed, the parent perovskite compound of the CMR phases, $LaMnO_3$, has been much studied on account of an isosymmetric phase transition from orthorhombic to pseudo cubic symmetry above 750 K. This transition is broadly understood as arising from the disordering of JT long axes along all three pseudo cubic directions[21,22]. However, in this case, the transition between these

two states is discontinuous and first order in nature leading to a phase coexistence. This discontinuity has been assumed to naturally occur since there is believed to be no single order parameter that can continuously describe the evolution from long-range OO to an orbital disordered phase in cases like this[23]. However, what we have shown here is at the $\frac{3}{8}$ doping level a coupling between two normal lattice modes and strain may form such a continuous OP. The strong coupling suggests that strain may effectively be used as a control parameter at the metal–insulator phase transition. Indeed, coherent compressive tetragonal strain fields in $La_{\frac{5}{8}-y}Pr_yCa_{\frac{3}{8}}MnO_3$, $y = 0.425$ generated via growth on $LaAlO_3$ substrate, providing a tetragonal (1-long 2-short)-type strain, as we have observed here, have been shown to stabilise the insulating state[24]. Thus our observation of the striped OO–CD with a clear tetragonal elongation of the lattice parameters provides a microscopic-based mechanism for models which have been used in describing the phenomenology of the strain induced transitions in these systems[25].

The $x < \frac{1}{2}$ state in the manganite system has been the subject of many theoretical studies[26]. However, these have considered only fully ordered models in which 2 out of 3 or 3 out of 4 B-sites are expected to have OO, so incompatible with $x$ precisely equal to $\frac{3}{8}$. The formally $Mn^{3+}$ sites in these models have alternating directions of their JT long axes with respect to pseudo cubic $a$ and $b$, and so are clearly inconsistent with the metric distortions observed in our data. On the other hand, we find that our experimentally determined OO–CD structure is robust with respect to relaxation under density functional theory (DFT) + $U$. So as not to unduly bias our calculations towards a particular solution, we select one global $U$ and $J$ that we apply across all Mn sites. The value chosen of $U = 0.5$ and $J = 0.0$ represents a compromise between higher values (e.g. $U = 2–4$ eV) required to open up a gap at the Fermi level for a pure insulator solution (full orbital and charge order) and values ($U = 0$) appropriate for an itinerant state (full CD). Within this approximation, the relaxed $\Sigma_2$ amplitude is in very good agreement with the experiment, while $M_3^+$ is somewhat reduced leading to a 2:1 ratio (see Supplementary Table 3 for full comparison of mode amplitude). Qualitatively similar results are obtained for a wide range of $U$ and $J$. The reduced mode ratio means that the electron density in the OO layers bleeds out slightly more into the CD than implied experimentally. The origin of these small quantitative disagreements could be due to the fact we consider only one average $U$ and $J$ for all Mn sites, and that we do not model the experimentally observed spin canting and additional incommensurability within the present level of theory. Nevertheless, the microscopic picture of layers with distinct OO (Mn with JT long axes) and CD, is clearly maintained as evident by the Jahn-Teller long axis of formally $Mn^{3+}$ (Fig. 4a) and spin density isosurfaces plotted for energies just below the Fermi level in Fig. 4c. Near-complete spin polarisation is evident in the OO $Mn^{3+}$ layer at fractional coordinate $z = 0$ (Fig. 4d) with the majority of electron density being captured by an isosurface consistent with the angular distribution function of a $d_{z^2}$-type orbital. $Mn^B$ sites at $z = \frac{1}{4}, \frac{1}{2}$ and $\frac{3}{4}$ have diminished levels of spin polarisations and more isotropic isosurfaces. Since the degeneracy of the $e_g$ orbitals are lifted by the JT distortion, the layers at $z = 0$ are expecting to be effectively insulating. In fact, the Fermi surface (Fig. 4d), consists of just 4 sheets that are perpendicular to $\mathbf{k}_z$, indicate that conductivity should be confined to one-dimensional chains along $b$.

Next we turn our attention to the simple perovskite systems in which the maximum CMR effect is observed. As a representative family member we take here $La_{\frac{5}{8}-y}Pr_yCa_{\frac{3}{8}}MnO_3$, $y = 0.375$ (LPCMO hereafter) which is found to have one of the largest

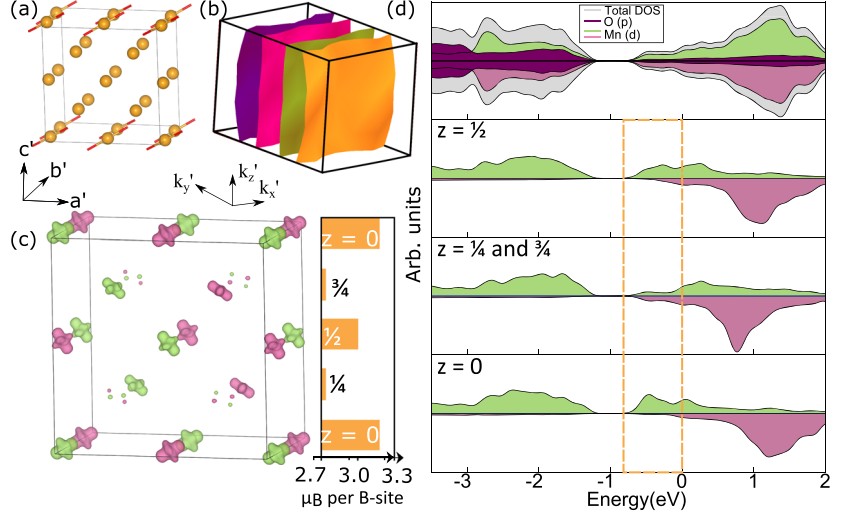

**Fig. 4 DFT + U results of the structural relaxation of the OO–CD model for the $x = \frac{3}{8}$ prototype system. a** The setting used in the DFT calculation is $\mathbf{a}' = \frac{\mathbf{a}}{2} + \frac{\mathbf{c}}{2}, \mathbf{c}' = \frac{\mathbf{a}}{2} - \frac{\mathbf{c}}{2}$ and $\mathbf{b}' = \mathbf{b}$ with respect to the experimental setting. Only the $Mn^B$ sublattice, with Jahn-Teller long bonds, is shown for clarity. **b** The Fermi surface consists of sheets perpendicular to $\mathbf{k}'_y$ indicative of one-dimensional conductivity. **c** Magnetic moments in Bohr magnetons ($\mu_B$) for $Mn^B$ by layer and magnetisation density isosurface plots at value of ±0.03 (positive green, negative pink) for energy window 0 to −0.8 eV below the Fermi level, as indicated in the site- and spin-resolved density of states (DOS) (**d**). Only the projected DOS for the spin-up Mn sites are shown in the various layers.

CMR responses[2]. We prepare this compound using the same high-pressure techniques that allowed us to synthesise the highly crystalline 134 prototype samples. High-resolution XRD studies reveal that the microstrain is intrinsically about three times higher ($e_0 = 0.04\%$) than in our prototype system, while the macrostrain is about three times lower (0.18%, see Supplementary Fig. 10 for comparison). At 130 K, near the metal-to-insulator phase transition, our OO–CD model for the prototype system provides a fair description of superstructure diffraction peaks (Supplementary Fig. 11). A refinement of the lattice mode amplitudes yields a 1:1 ratio of $\Sigma_2 : M_3^+$. The amplitudes are reduced by approximately a factor of 2 with respect to the prototype systems. The reduced amplitudes likely arise from the phase fraction of the OO–CD phase being significantly less than unity, such as is expected at the boundary between insulator and metallic states in these systems. Both increased microstrain, pseudo-cubic symmetry and additional degrees of freedom on account of the *Pnma* tilt structure (in which distortions transforming as $M_3^+$ are already permissible by symmetry) conspire against a more robust refinement. All reasons which underpin why this has remained an unsolved problem for so long. Again we find that our OO–CD solution is robust with respect to relaxation under DFT + U for the LPCMO phase, with $\Sigma_2 : M_3^+$ in a 2:1 ratio and elongated Mn–O bond distances (2.07 Å) in 1 in 4 layers (see Supplementary Table 3), and the spin-polarised isosurfaces (Supplementary Fig. 12) and density of states (Supplementary Fig. 13) being qualitatively very similar to those of our prototype system.

Finally, we explore the coupling between the OO–CD order and magnetism that is crucial in mediating the MR effect. Our neutron-diffraction data (Fig. 1a and Supplementary Fig. 14) show that the well-known CE-type antiferromagnetic (AFM) order, which is intrinsically coupled to the OO state, is also observed in our half-doped prototype system (A = Na). There is a steady decrease in this order (see Supplementary Fig. 14) with the frustration of the magnetic interactions resolving itself in an incommensurate modulation towards $x = \frac{3}{8}$. This incommensurate magnetic phase has recently been shown to be linked to the pCE-type magnetic ordering[27] known in the LPCMO systems[3], which itself is believed to compete with the FM ground state.

Hence, application of a magnetic field might be expected to remove this frustration in favour of an FM state. To test if such an FM state would disrupt our OO–CD model, leading to a fully three-dimensional metallic phase, we perform a DFT geometry optimisation of our LPCMO OO–CD structure but now with all Mn spins co-aligned. This results in a complete washing out of all signature of the modes associated with the OO (see Supplementary Table 3). The Mn–O bond distance now all fall within the range 1.946–1.953 Å precluding any charge or orbital order, and multiple, highly dispersive spin-polarised band crossing the Fermi energy are now evident (see Supplementary Fig. 15). Hence our OO–CD model is shown to be strongly coupled to the spin ordering in the structure, proving a natural mechanism through which the metallic FM state can emerge in the canonical CMR system under applied magnetic fields. However, due to the large octahedral tilt angles in our prototype system, which are effectively locked in place by the cation ordering, we do not expect these to exhibit CMR within an experimentally achievable magnetic field strength.

In conclusion, we have shown that $AMn_3^{A'}Mn_4^BO_{12}$ A = $Na_{1-x}Ca_x$ and $La_{1-x}Ca_x$ can act as a prototype system for canonical CMR perovskites, having the same electronic orderings observed in $LaMnO_3$ at zero and half-doped levels, and through the correspondences observed in, DFT relaxations and their diffraction data. Detailed crystallographic investigations of this prototype system have allowed us to identify a new kind of orbital order at the $x = \frac{3}{8}$-doped level of the manganite phase diagram consisting of OO and CD stripes. Our model of the evolution of the orbital and charge order is highly consistent with the macroscopic strain evolution across the phase diagram, and is robust with respect to relaxation under DFT + U for the both prototype and the canonical CMR system. The partially charged and OO phases at $x = \frac{3}{8}$, where a maximum in the MR effect is observed in related manganites, provides the missing intermediate between CE-type insulating phase and FM metallic phase. While CMR physics is undoubtedly linked to phase segregation in canonical LCMO and LPCMO systems, a better understanding of how to stabilise the order parameters that are entwined with OO and CD, will ultimately allow for a more precise control over a host of related physical phenomena.

## Methods

**Experimental details, data collection and symmetry analysis.** Polycrystalline $AMn_7O_{12}$ series and $La_{\frac{5}{8}-y}Pr_yCa_{\frac{3}{8}}MnO_3$, $y = 0.375$ (LPCMO) samples were prepared by solid-state reaction under high-pressure and high-temperature conditions. Stoichiometric amounts of $La_2O_3$ (Alfa Aesar 99.999%, preheat before use), $Na_2O_2$ (Sigma-Aldrich 97%), CaO (preheat from $CaCO_3$, Alfa Aesar 99.999%), $Pr_6O_{11}$ (Alfa Aesar 99.996%), $MnO_2$ (Alfa Aesar 99.997%) and $Mn_2O_3$ (Aldrich 99.99%) were well mixed and sealed in a platinum capsule. The capsule, boron nitrite insulating layer and graphite heater were assembled in a pyrophyllite cell and placed in a DIA-type cubic anvil high-pressure apparatus. The samples were treated at 8 GPa and 1600 K for 30 min then released to ambient condition.

Each synthesis produced ≈0.1 g sample. Judging by the X-ray powder diffraction and magnetic susceptibility, up to three samples where combined for neutron power diffraction. Uncombined samples where used for the Synchrotron X-ray powder diffraction (SXRD) experiments. Magnetic property measurements were carried out with a commercial magnetometer Quantum Design Magnetic Properties Measurement System (VSM-MPMS). Zero field cooled and field cooled magnetic susceptibility were measured at 5–300 K in an external magnetic field of 10 kOe.

Full details of the refined average structures are given in Supplementary Tables 1 and 2, Supplementary Fig. 5 and in Supplementary Data 1. SXRD experiments were carried out for crystal structure analysis. The powder samples were packed in a 0.1 mm borosilicate capillaries, to minimise the absorption effect, and measured with a 40 keV beam using the nine crystal multi-analyser detector at the high-resolution powder diffraction beamline ID22, ESRF. Samples were measured as a function of temperature at 700 K down to 10 K. Further temperature-dependent measurements were performed with 20 and 15 keV beam energy and a position sensitive MYTHEN detector at beamline 19A and 09A, Taiwan Photon Source and I11, Diamond Light Source, UK. Neutron powder diffraction data on the combined ≈0.3 g polycrystalline sample in cylindrical aluminium foil parcels with diameter 2 mm were carried out at beamline WOMBAT[28] ANSTO, Australia with a neutron wavelengths of ≈2.42 Å. Heat maps from this data have been created as a function of composition and $d$-spacing showing the magnetic scattering (see Supplementary Figs. 2 and 14). Samples of composition $A = Na_{0.4}Ca_{0.6}$, $Na_{0.9}Ca_{0.1}$ and $La_{0.8}Ca_{0.2}$ were measured at the time-of-flight powder diffraction beamline WISH, ISIS, down to 2 K allowing for a robust assignment of magnetic orderings in these parts of the phase diagram. The obtained data were analysed within the Rietveld method using the program TOPAS, and refinements were done using the symmetry adapted displacements formalism as parameterised by ISODISTORT[29] and implemented through the Jedit interface with TOPAS[30]. Atomic displacements, as well as lattice strain and commensurate magnetic orderings were all analysed within this framework. In order to facilitate an easy comparisons with literature results on other simple perovskite manganites, the parameterisations were all performed with respect to a hypothetical aristotype $A_{0.25}Mn_{0.75}MnO_3$ with $Pm\bar{3}m$ symmetry with A at (0,0,0); $Mn^B$ at (0.5, 0.5, 0.5) and O at (0.5, 0.5, 0). From this starting point, ISODISTORT was used to generate a model in P2 symmetry (basis = [(4,0,0),(0,2,0),(0,0,4)], origin = (0,0,0),) that captures all of the necessary symmetry breaking, whilst retaining the high-symmetry lattice directions. Appropriate symmetry constraints have been used to reflect the true structural symmetry, and models in both this setting and those of the space group $C2/c$ are given as part of the SI. Mode amplitudes reported in this manuscript are in Å and represent the square root of the sum of the square of all atomic displacements associated with a given mode within the primitive cubic $Pm\bar{3}m$ perovskite unit cell ($A_p$ values as defined in ISODISTORT).

Single-crystal diffraction experiments were performed on $A = Na_{0.4}Ca_{0.6}$ in Experimental Hutch 1 of I19, Diamond Light Source using a Pilatus 2M Detector and a wavelength of 0.68890 Å. Crystal orientation was determined using XDS[31] and reciprocal space reconstructions performed using a in-house script called Meerkat. The reconstructions (Supplementary Fig. 6) clearly show that the propagation vectors have been correctly identified at opium doping level, and that no structured diffuse scattering is present, indicating that the phase is well described by the average crystallographic model. The measured single-crystal data could not be used for refinement for two reasons. Firstly, since the data collection was optimised for weak superstructure reflections, strong Bragg reflections were over-saturated and could not be integrated. Secondly, the crystal showed merohedral twinning with six near-identically proportioned domains.

The irrep. labels given throughout this paper are with respect to aristotype $ABO_3$ perovskite with the crystallographic setting defined above. Within this framework, the A-A' cation ordering, transforms as the irrep. $M_1^+$(a;a;a), and the in-phase octahedral rotational displacements, which transform as $M_2^+$(a;a;a)[12,32], are additional distortions already present in the $Im\bar{3}$ 134 parent structure. These orderings on their own do not couple to any symmetry-breaking strain. The orbital ordering described here transform as irrep. $M_3^+$(0;a;0) $\oplus$ $\Sigma_2$(0,0;0,0;a,a;0,0;0,0;0,0) with active propagation vector $\mathbf{k} = (1/2,0,1/2)$ and $(1/4,0,1/4)$, respectively. It is the evolution of these order parameters coupling to the strain that leads to the observed metric symmetries. The symmetrised tetragonal and orthorhombic strains in the OO structures transform as the doubly degenerate representation $\Gamma_3^+(a,b)$ where the special cases $b = 0$, $a = b/\sqrt{3}$ or $a = -b/\sqrt{3}$ correspond to a purely tetragonal-type distortions, with unique $c$, $b$ and $a$ axis of the setting used for the refinements.

In our phase diagram, pseudo-tetragonal states are observed at the $x = \frac{1}{2}$ ($b$-unique) and $x = \frac{3}{8}$ ($c$-unique), and hence plotting $\Gamma_3^+(b)$ / $\Gamma_3^+(\sqrt{3}a)$, which we define as $t$ in the main text, provides a measure of the continuously varying lattice distortion between values $t = 1$ and $t = 0$, respectively. Individually, the primary order parameters (POPs) of the OO, induce secondary strain order parameters (SOPs). In the case of $M_3^+$(0;a;0) this is the strain transforming as $\Gamma_3^+(\frac{a}{2},\frac{\sqrt{3}a}{2})$ (unique $b$ short axis of parent) and for $\Sigma_2$(0,0;0,0;a,0;0,0;0,0;0,0) as $\Gamma_3^+(\frac{a}{2},-\frac{\sqrt{3}a}{2})$ (unique $a$ axis short). Note the importance of order parameter directions with respect to each other. The amplitude of these SOPs is driven by the POPs that describe the orbital OO. Assuming equal coupling strengths, the point at which the amplitudes of POP $M_3^+$ and $\Sigma_2$ approach equality at $x = \frac{3}{8}$ will lead to an effective strain $\Gamma_3^+(a,0)$ corresponding to a long $c$-unique axis as observed experimentally (Fig. 2). The magnitude of the strain is somewhat reduced due to increased CD at $x = \frac{3}{8}$. The sheer, monoclinic and rhombohedral-type strains transform as $\Gamma_5^+(0,0,a)$ and $\Gamma_5^+(a,a,a)$, respectively. These strains seem to scale with amplitudes of $\Sigma_2$ - $M_3^+$ across the whole system passing through zero at or near $\frac{3}{8}$ (see Supplementary Fig. 7). Hence, metrically tetragonal symmetry appears to be exact for $A = Na_{0.4}Ca_{0.6}$, within the high resolution (±0.003 Å, 0.025%, see Supplementary Fig. 9) of the experiment. The temperature evolution of these lattice parameters (see Supplementary Fig. 8) show that metric tetragonal state is not accidental, with lattice parameters locked-in to a pseudo tetragonal state below the OO–CD phase transition temperatures. However, it is important to realise that despite this, on account of the 1:3 cation ordering and octahedral rotations, which break the 4-fold axes of the aristotype, the highest symmetry subgroup that this system could adopt is orthorhombic. In fact, the coupling with physically meaningful distortions related to the orbital ordering such as $M_3^+$ and $\Sigma_2$ yield monoclinic symmetry. It can hence only be an exact cancellation of these strain terms in the Landau style expansion of the free energy, which appear to occur as the magnitude of OP $M_3^+$ and $\Sigma_2$ approach each other, which can produce this metric symmetry. On the other hand, symmetry analysis of the $Pnma$ perovskite structure that describes the octahedral tilting in the LCMO family of related perovskites reveals strain transforming as $\Gamma_3^+(\frac{a}{2},\frac{\sqrt{3}a}{2})$ (respecting our choice of in phase octahedral rotation transforming as $M_2^+$(0;a;0)), and sheer strain (w.r.t. $Pm\bar{3}m$ setting) as $\Gamma_5^+(0,0,a)$, even in the absence of OO. Hence in these systems, the strains do not provide a straightforward insight into the evolution of the OO across the doping series. Put another way, tilting of the octahedra will have a pronounced effect on how the JT distortions couple to the lattice distortions. Furthermore, unlike in our prototype system, atomic displacements associate with the $M_3^+$ OO are already induced as SOPs through coupling between the octahedral tilts, meaning that a refined amplitude may not necessary be taken as being indicative of an associated electronic instability.

**The solid solution.** In the present study, our aim is to hole-dope the system (with respect to $A = La$) in a similar manner to that which has been used in the study of the various LCMO related materials. The added complication of having transition metals on both the perovskite A and B site with potentially variable oxidation states means that we must first confirm that the average change in valence electrons is associated with the desired $Mn^B$ site only. To do this we have performed Rietveld refinements of the structures against high-resolution X-ray diffraction data collected at 700 K, a temperature that is above all structural phase transition. In $Im\bar{3}$, the structure, and hence bond lengths, angles and BVS values, are determined exclusively by the refinement of a single lattice and two oxygen displacement parameters, and hence these values are robustly determined. Figure 1c shows that while the A-site BVS is constant at about 2.9+ across the series, near the expected 3+ value based on its square planar coordination, the B-site BVS varies smoothly from 3.05 at the La-rich end to 3.55+ at Na-rich end. This corresponds to the expected half electron doping across the phase diagram on the B-site, consistent with the formal $Mn^B$ valence assigned on the $x$-axis of Fig. 1c. We hence refer to regions of this phase diagram using the formal $Mn^B$ valence.

Figure 1c shows two slightly different slope in volume changes centred on middle member $CaMn_7O_{12}$ arising from the small difference in Na(1.39 Å) and La(1.36 Å) radii with respect to Ca (1.34 Å). However, the general trend shows that the vast major of the change is associated with the change in valence. The change in the octahedral rotation angle Fig. 1c varies smoothly from 137.5° to 140° across the series. The temperature-dependent change across the series also leads to a change of less than 1.5°, and there are no soft-mode instabilities in these 134 perovskites. In this respect, our solid solution is best contrasted with the electron-rich end of the narrow-bandwidth manganites (e.g. $LaMnO_3$ to $La_{0.5}Ca_{0.5}MnO_3$). However, for our 134 systems, the small change to the bond angle in what are already very large deviations from 180°, preclude any significant further narrowing as a function of $x$ or temperature such as complicates the phase diagram of many narrow-bandwidth manganites. In these respects our phase diagram can be view as a prototype for studying the effect of electron doping on orbital ordering.

**Details of DFT calculations.** Density functional theory (DFT) calculations were performed on the manganite systems using a projector-augmented wave method[33] as implemented within the Vienna ab initio simulation package (VASP)[34,35]. We

used the PBEsol[36] + $U$ framework as implemented by Lichtenstein et al.[37], with $U = 0.5$ eV and $J = 0$ eV. While the JT active $Mn^{3+}$ end-member is known to require a sizable $U$ to reproduce experimental features, it has been shown that the hole-doped manganites do not[38,39]. A plane wave cutoff of 1000 eV and a $3 \times 4 \times 3$ k-point mesh (for the $2\sqrt{2} \times 2 \times 2\sqrt{2}$ 80-atom cell) was employed. Atomic positions were relaxed (to forces below 5 meV/Ang) starting from the experimentally determined structure, while the lattice vectors were constrained to the experimental. A collinear antiferromagnetically ordered spin structure for the Mn sites was generated from the experimentally determined configuration (Supplementary Fig. 14), with the smaller canting along **c** ignored in this approximation. This relaxed structure was compared with a relaxation where a ferromagnetically ordered structure had been imposed. Both the 134 prototype and LPCMO perovskite were considered. To account for the hole doping, we removed $\frac{1^{th}}{8}$ electron per Mn B-site in the valence of $CaMn_3Mn_4O_{12}$, and $\frac{3^{th}}{8}$ electron per Mn site in the valence of $PrMnO_3$. In addition to the four Fermi surface sheets in Fig. 4b, created by two dispersive bands (each crossing at $\pm\mathbf{k}$), a small Fermi surface pocket around $\Gamma$ also exists from a third non-dispersive band. However, this third non-dispersive band only just crosses the Fermi level (Supplementary Fig. 15) and hence this pocket around $\Gamma$ would quite likely disappear with slight changes to the calculation (e.g. atomic structure, XC functional or Hubbard $U$ value).

## Data availability

The experimental diffraction data generated in this study are deposited in Figshare with the identifier 10.6084/m9.figshare.14823678. For the computational work, authors can confirm that all relevant data are included in the paper and its supplementary information files.

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

## Acknowledgements

This work was supported by a joint Royal Society and Ministry of Science and Technology (MOST), Taiwan International Exchange Scheme (IE141335, 104-2911-I-002-535 and 105-2911-I-002-515). The synchrotron beam time used in this paper was at ID22, European Synchrotron Radiation Facility (HC2331), I11 through the Diamond Light Source Block Allocation Group award "Oxford-Warwick Solid State Chemistry BAG to probe composition-structure-property relationships in solids" (EE18786) and Taiwan Photon Source (2016B0091 and 2017-3-298). Neutron powder diffraction was performed at Wombat (P4864 and P5877) and at WISH, ISIS (RB1820434). Single-crystal diffraction was performed at I19, Diamond Light Source under proposal CY27647. M.S.S. would like to acknowledge the Royal Commission for the Exhibition of 1851 and the Royal Society (UF160265) for fellowships, W.T.C. would like to acknowledge MOST, Taiwan for financial support (103-2112-M-002-022-MY3), and N.C.B would like to acknowledge the UK Materials and Molecular Modelling Hub for computational resources (partially funded by the EPSRC project EP/P020194/1). A.S. is funded by the Swiss National Science foundation (grant PZ00P2-180035). The authors would like to acknowledge staff at beam lines ID22, WISH and I19 for their assistance.

## Author contributions

The study was designed by M.S.S. and W.T.C. Sample synthesis and physical property characterisation was performed by W.T.C. with assistance from C.C.C.; W.T.C., M.S.S., A.S., Y.C.C. and C.W.W. performed the diffraction experiments, with M.S.S. leading on the crystallographic analysis of the data. The DFT calculations were performed by N.C.B. The paper was written by M.S.S. with contributions from all other co-authors.

## Competing interests

The authors declare no competing interests.
