## [Peer Review File. · Nature Communications]

REVIEWER COMMENTS

Reviewer #1 (Remarks to the Author):

A systematic structural study on a series of perovskite manganese oxide compounds $\text{AMn}_7\text{O}_{12}$ is reported in this manuscript. High-quality neutron and synchrotron x-ray diffraction data are summarized in the structure phase diagram as a function of temperature and Mn average valence.

It is a beautiful and interesting study on one hand. The originality is high enough, the data analysis is approvable, and an orbital order that has not been reported so far is found. On the other hand, the proposed connection with the so-called colossal magnetoresistivity in many perovskite manganese oxide compounds may not be acceptable. I also say that the presentation of this manuscript would be a bit unfriendly. In the following I raise some specific examples.

1.

Previous neutron scattering studies showed that many perovskite manganese oxide compounds with the hole doping of $3/8$ host a so-called CE-type antiferromagnetic order. The magnetic order is characterized by two modulation vectors $(1/4 \ 1/4 \ 1/2)$ and $(1/2 \ 0 \ 1/2)$ in the primitive cubic perovskite. What kind of magnetic order did the authors find for the formal Mn valence of about $3/8$? Although a neutron diffraction pattern is provided in the supplementary material, I do not understand the arrangement of the spins.

2.

Are the obtained neutron diffraction data consistent with their DFT+U calculation?

3.

The discussion about the relation between the obtained experimental data and CMR effect is rather confusing. For example, the magnetic field effect is discussed based only on the DFT+U calculation and does not seem to be supported by the experimental result. Why are no experimental data of $\text{AMn}_7\text{O}_{12}$ in a magnetic field provided in this manuscript?

4.

It is rather difficult to understand the orbital order in the rhombohedral phase, shown in Fig. 2(d). The unit cell contains two Mn^{4+} , two Mn^{3+} with the orbital of x^2-y^2 , two Mn^{3+} with y^2-z^2 , and two Mn^{3+} with z^2-x^2 ?

5.

Do all the compounds exhibit a metal-insulator transition at the orbital ordering temperature?

6.

I would like to recommend that the lattice parameters including angles in the rhombohedral and monoclinic phases should be provided in the main article.

7.

I do not understand Figure 4(d). It does not seem that the top panel agree with the other three panels. Is the system in the spin polarized state but not an antiferromagnetic state?

8.

I would recommend that the English should be improved.

Reviewer #2 (Remarks to the Author):

In order to understand the behavior of canonical systems, $R_{1-x}\text{Ca}_x\text{MnO}_3$, near the optimal doping level, the authors investigated other systems, namely, $\text{La}_{1-x}\text{Ca}_x\text{Mn}_7\text{O}_{12}$ and $\text{Na}_{1-x}\text{Ca}_x\text{Mn}_7\text{O}_{12}$. $\text{La}_{1-x}\text{Ca}_x\text{Mn}_7\text{O}_{12}$ and $\text{Na}_{1-x}\text{Ca}_x\text{Mn}_7\text{O}_{12}$ solid solutions allow variations of the average oxidation state of Mn at the B site in wide ranges while keeping other distortions nearly constant due to the specific crystal structure. They present very detailed results on 21 samples – this is huge work. They obtained good understanding of the structural behavior and tiny structural features related to $R_{1-x}\text{Ca}_x\text{MnO}_3$ and $\text{La}_{1-x}\text{Ca}_x\text{Mn}_7\text{O}_{12}$ and $\text{Na}_{1-x}\text{Ca}_x\text{Mn}_7\text{O}_{12}$. Therefore, I think that this paper should be published after minor revision. The comments are listed below.

1. The authors have errors in general formulae. For example, $\text{Na}_{x-1}\text{Ca}_x$ and $\text{La}_{x-1}\text{Ca}_x$: the sum should be 1 – it is $2x-1$. $\text{La}_{3/8-y}\text{Pr}_y\text{Ca}_{3/8}\text{MnO}_3$: the sum should be 1 – it is $6/8$.
2. I think that for the convenience of readers the abstraction should mention chemical formulae, not just saying “a prototype system” and “a canonical system”.
3. The authors collected high-quality neutron and synchrotron xrd data. Did the authors see any incommensurately modulated reflections near pure $\text{CaMn}_7\text{O}_{12}$? In addition to a rhom-cubic transition (as shown on Figure 2c) $\text{CaMn}_7\text{O}_{12}$ shows an additional structural transition at lower temperature. I think this transition should be shown on the phase diagram.
4. Figure 1 should show diffraction data at different temperatures. Therefore, every panel should specify temperature. In the current presentation only the top-left and bottom-right panels give temperatures. $\text{Na}_{0.9}\text{Ca}_{0.1}\text{Mn}_7\text{O}_{12}$ is cubic at room temperature (according to Figure 2c). Then why did the authors show diffraction data at 700 K and not at RT? I think that the inset with the crystal structure in Figure 1a is too small to illustrate something and should be removed.
5. The title of ref. 6 is not full.
6. Figure S8 shows temperature dependence of the lattice parameters. Monoclinic angles should also be shown for the samples with monoclinic symmetry.

REVIEWER COMMENTS

Reviewer #1 (Remarks to the Author):

A systematic structural study on a series of perovskite manganese oxide compounds AMn_7O_{12} is reported in this manuscript. High-quality neutron and synchrotron x-ray diffraction data are summarized in the structure phase diagram as a function of temperature and Mn average valence.

It is a beautiful and interesting study on one hand. The originality is high enough, the data analysis is approvable, and an orbital order that has not been reported so far is found. On the other hand, the proposed connection with the so-called colossal magnetoresistivity in many perovskite manganese oxide compounds may not be acceptable. I also say that the presentation of this manuscript would be a bit unfriendly. In the following I raise some specific examples.

1.

Previous neutron scattering studies showed that many perovskite manganese oxide compounds with the hole doping of $3/8$ host a so-called CE-type antiferromagnetic order. The magnetic order is characterized by two modulation vectors $(1/4\ 1/4\ 1/2)$ and $(1/2\ 0\ 1/2)$ in the primitive cubic perovskite. What kind of magnetic order did the authors find for the formal Mn valence of about $3/8$? Although a neutron diffraction pattern is provided in the supplementary material, I do not understand the arrangement of the spins.

The commensurate part of the magnetic structure is consistent with the CE-type antiferromagnetic order with propagation vectors $(1/4\ 1/4\ 1/2)$ and $(1/2\ 0\ 1/2)$. And additional incommensurate modulation of this, as discuss by Johnson et al. (our reference [27], Physical Review Letters 120, 257202 (2018)) is present as appreciable from the heat map plot in Fig S14 at ~ 4.5 A. Johnson et al. show this phase to be analogous to the pCE-type phase discuss in the many perovskite manganites. The difference between the pCE and incommensurability possibly arises due to frustration between ordering on the A and B site manganite ions.

We amend the appropriate section:

Our neutron diffraction data (Fig. \ref{structure}(a) and Fig. S14), shows that the well know CE-type antiferromagnetic order, that is intrinsically coupled to the orbital ordered state, is also observed in our half-doped prototype system ($A = Na$). There is a steady decrease in this order (see Fig. S14) with the frustration of the magnetic interactions resolving itself in an incommensurate modulation towards $x = 3/8$.

2.

Are the obtained neutron diffraction data consistent with their DFT+U calculation?

Periodic boundary conditions of the DFT can only accommodate the commensurate part of the magnetic structure. This is found to be consistent with the commensurate part of the model derived from the neutron diffraction.

We amend the following statement to clarify this point:

A collinear antiferromagnetically ordered spin structure for the Mn sites was generated from the experimentally determined configuration of the prototype (Fig. S14), and it was compared with a relaxation where a ferromagnetically ordered state had been imposed. The relaxed AFM spin structure is consistent with the experimentally observed CE-type magnetic ordering, but the additional incommensurate modulation cannot be modelled within the periodic boundary conditions of DFT.

3.

The discussion about the relation between the obtained experimental data and CMR effect is rather confusing. For example, the magnetic field effect is discussed based only on the DFT+U calculation and does not seem to be supported by the experimental result. Why are no experimental data of AMn7O12 in a magnetic field provided in this manuscript?

We acknowledge fully that our 134 prototype system may not exhibit CMR as it is unlikely to show the perquisite phase coexistence known to be intrinsic to this phenomenon. It is however this very same phase coexistence that has precluded a detailed study of the OO structure at the 3/8th doping level in the canonical systems, hence the need for our prototype systems. We have established that the model derived from the detailed structural work on our prototype system describes the OO/AFM state of the canonical LPCMO system well. The magnetic properties and field induced phase transitions are well studied in LPCMO. Hence our DFT study on LPCMO using our novel OO-CD structure provides a convenient way to illustrate the coupling between magnetic and charge and orbital degrees of freedom in these systems and is a very valuable endeavour.

Investigating if this can be experimentally achieved by applications of large external magnetic fields in the prototype system, is of course also a very interesting research question and will form a substantial piece of future research. However, the narrow band gap nature of the prototype system and large octahedral tilt angles, that are effectively lock in place by the cation ordering, probably mean that the magnitude of required magnetic fields is well beyond experimental reach.

We amend the ultimate sentence in the penultimate paragraph to the following statements to further clarify this point.

“Hence our OO-CD model is shown to be strongly coupled to the spin ordering in the structure, proving a natural mechanism through which the metallic ferromagnetic state can emerge in the canonical CMR system under applied magnetic fields. However, due to the large octahedral tilt angles in our prototype system, that are effectively locked in place by the cation ordering, we do not expect these to exhibit CMR within an experimentally achievable magnetic field strength. “

4.

It is rather difficult to understand the orbital order in the rhombohedral phase, shown in Fig. 2(d). The unit cell contains two Mn⁴⁺, two Mn³⁺ with the orbital of x²-y², two Mn³⁺ with y²-z², and two Mn³⁺ with z²-x²?

We actually removed text pertaining to this from a previous draft to meet the word limit requirement for Nature Physics. We now reinsert this:

A = Ca (Mn_B^{3.25+}), in which OO has previously been reported \cite{Bochu1980} in a rhombohedral structure, may also be viewed as C-type orbital (and charge order) but now with the planes of OO perpendicular to [1 1 1], preserving a three fold axis of the cubic aristotype. It

contains three JT active Mn^{3+} sites, which have 4 long: 2 short bonds, for every non-JT active Mn^{4+} site (Fig. \ref{Fig2} (d)). The apparent 4-long 2-short distortion is actually due to an averaging of disordered JT 2-long 4-short bonds about the [111] axis \cite{Streltsov2014} which resolves itself in long range incommensurate order at 250 K in $A = \text{Ca}$ \cite{Sawinski2009,Perks2012}, that we find is washed out rapidly with doping (Fig. \ref{Fig2} (c)). This rhombohedral phase persists up to a doping levels of $\text{Mn}_B^{3.325+}$ ($A = \text{Ca}_{0.7}\text{Na}_{0.3}$) beyond which point a pronounced change in lattice symmetry occurs.

5.

Do all the compounds exhibit a metal-insulator transition at the orbital ordering temperature?

Yes, in general one can see an anomalous step in the resistivity at the OO-CD temperature indicative of electron localisation (see below). Again, for reason given in the manuscript and reply to point 3, we do not want to focus on the resistivity of our prototype samples, by construction these are narrow band manganites with large magnitudes of octahedral tilts that are locked in place by the cation ordering.

6.

I would like to recommend that the lattice parameters including angles in the rhombohedral and monoclinic phases should be provided in the main article.

Thank you for this suggest, we have now added these to figure 2.

7.

I do not understand Figure 4(d). It does not seem that the top panel agree with the other three panels. Is the system in the spin polarized state but not an antiferromagnetic state?

We thank the referee for highlighting this confusion. The system is in the AFM state. The caption failed to specify that the three bottom panels are plotting the projected DOS only for the spin-up Mn sites within each of the three layers. This differentiation is necessary to highlight the origin of the depleted spin density on individual Mn sites within the $z=1/4$ and $3/4$ layers (as seen in the central panel). We have now adjusted the figure caption accordingly.

8.

I would recommend that the English should be improved.

The manuscript has been written and proofread by two native English speakers. Some minor typographic errors have been corrected on this revision.

Reviewer #2 (Remarks to the Author):

In order to understand the behavior of canonical systems, $R_{1-x}Ca_xMnO_3$, near the optimal doping level, the authors investigated other systems, namely, $La_{1-x}Ca_xMn_7O_{12}$ and $Na_{1-x}Ca_xMn_7O_{12}$. $La_{1-x}Ca_xMn_7O_{12}$ and $Na_{1-x}Ca_xMn_7O_{12}$ solid solutions allow variations of the average oxidation state of Mn at the B site in wide ranges while keeping other distortions nearly constant due to the specific crystal structure. They present very detailed results on 21 samples – this is huge work. They obtained good understanding of the structural behavior and tiny structural features related to $R_{1-x}Ca_xMnO_3$ and $La_{1-x}Ca_xMn_7O_{12}$ and $Na_{1-x}Ca_xMn_7O_{12}$. Therefore, I think that this paper should be published after minor revision. The comments are listed below.

1. The authors have errors in general formulae. For example, $Na_{x-1}Ca_x$ and $La_{x-1}Ca_x$: the sum should be 1 – it is $2x-1$. $La_{3/8-y}Pr_yCa_{3/8}MnO_3$: the sum should be 1 – it is $6/8$.

Many thanks for spotting this. We have fixed this to $Na_{1-x}Ca_x$ and $La_{5/8-y}Pr_yCa_{3/8}MnO_3$

2. I think that for the convenience of readers the abstraction should mention chemical formulae, not just saying “a prototype system” and “a canonical system”.

The abstract has been shortened to meet the journal style guidelines. The prototype systems chemical formula is now given.

3. The authors collected high-quality neutron and synchrotron xrd data. Did the authors see any incommensurately modulated reflections near pure $CaMn_7O_{12}$? In addition to a rhom-cubic transition (as shown on Figure 2c) $CaMn_7O_{12}$ shows an additional structural transition at lower temperature. I think this transition should be shown on the phase diagram.

The referee is quite right. There is an incommensurate phase transition below 250K, and satellite peaks are evident in pure CaMnO₁₂. However even at very lightly doped level (A = Ca_{0.9}Na_{0.1}) these are almost entirely suppressed. As this is not the main thrust of our article, we have not collected extensive temperature dependent data to investigate this. However, we add a region to the phase diagram to schematically represent the IC phase.

We have amended the text as follows:

$A = \text{Ca}(\text{Mn}_{B}^{3.25+})$, in which OO has previously been reported \cite{Bochu1980} in a rhombohedral structure, may also be viewed as C-type orbital (and charge order) but now with the planes of OO perpendicular to $[1\ 1\ 1]$, preserving a three fold axis of the cubic aristotype. It contains three JT active Mn^{3+} sites, which have 4 long: 2 short bonds, for every non-JT active Mn^{4+} site (Fig. \ref{Fig2} (d)). The apparent 4-long 2-short distortion is actually due to an averaging of disordered JT 2-long 4-short bonds about the $[111]$ axis \cite{Streltsov2014} which resolves itself in long range incommensurate order at 250 K in $A = \text{Ca}$ \cite{Sawinski2009,Perks2012}, that we find is washed out rapidly with doping (Fig. \ref{Fig2} (c)).

4. Figure 1 should show diffraction data at different temperatures. Therefore, every panel should specify temperature. In the current presentation only the top-left and bottom-right panels give temperatures. Na_{0.9}Ca_{0.1}Mn₇O₁₂ is cubic at room temperature (according to Figure 2c). Then why did the authors show diffraction data at 700 K and not at RT? I think that the inset with the crystal structure in Figure 1a is too small to illustrate something and should be removed.

We chose here 700 K since this is the temperature at which all compositions adopt cubic symmetry, but we could have equally plotted RT data. We address the ambiguity in our caption:

Diffraction data on AMn₇O₁₂, showing cubic $I\bar{m}\bar{3}$ Na_{0.9}Ca_{0.1}Mn₇O₁₂ (top left, 700 K), rhombohedral $R\bar{3}$ La_{0.4}Ca_{0.6}Mn₇O₁₂ (top right, 80 K), and monoclinic $I2/m$ La₁Mn₇O₁₂ (bottom left, 80 K) and pseudo tetragonal ($C2/m$) Na_{0.4}Ca_{0.6}Mn₇O₁₂ (bottom right, 80 K) with inset superstructure peaks.

Inset has been removed and we add another part to the figure showing an enlarged crystal structure.

5. The title of ref. 6 is not full.

Thanks for spotting this, this is now amended.

6. Figure S8 shows temperature dependence of the lattice parameters. Monoclinic angles should also be shown for the samples with monoclinic symmetry.

We have added angles to this figure as well as to Figure 2 in the main text.

REVIEWER COMMENTS

Reviewer #1 (Remarks to the Author):

I have investigated the authors' response to the comments of two reviewers. I still have some concerns as follows.

1. I gave a comment on the magnetic structure of the (Ca,Na) solid solution in my previous report. The authors respond that the system has two magnetic structures: the famous CE-type and an additional incommensurate magnetic structure. They revise a section as follows:
"Our neutron diffraction data (Fig. \ref{structure}(a) and Fig. S14), shows that the well know CE-type antiferromagnetic order, that is intrinsically coupled to the orbital ordered state, is also observed in our half-doped prototype system (A = Na). There is a steady decrease in this order (see Fig. S14) with the frustration of the magnetic interactions resolving itself in an incommensurate modulation towards $x = 3/8$."

I still do not understand how the neutron diffraction data were analyzed to pin down the magnetic structures. The neutron diffraction data are provided as heat maps both in the Figs. 2(a) and S14. I am afraid that such a heatmap is not suitable for understanding the magnetic structure. For example, 'a steady decrease in this order' is not clearly shown. I recommend that the raw diffraction patterns are shown in the conventional way with each peak indexed. In addition, the mS2 and mM2+ modes in Fig. S14 are not defined, which would be confusing. The schematics in the inset are also confusing due partly to the superposition of two antiferromagnetically coupled 'layers'. It is important because the incommensurate pseudo-CE-type structure proposed by Johnson et al. has ferromagnetic stacking for some sites while antiferromagnetic stacking for the other sites. In the AMn7O12 system, the Mn moments on A sites can also contribute to the magnetic order. In the inset, however, Mn moments on A sites is, perhaps, omitted. I do not understand what red lines attached to some Mn ions in the inset denote, either.

The correspondence between the magnetic structure and orbital order is not clear, either. If the authors suggest the coexistence of two or more magnetic phases in some compounds in some temperature ranges, multiple orbital states may also coexist.

2. I raised a question about the consistency between the neutron data and DFT+U calculation. The authors reply that the commensurate part was reproduced by DFT+U while the incommensurate part was not. I understand that the boundary condition for the calculation is harmful in the incommensurate case. They revise a section as follows:
"A collinear antiferromagnetically ordered spin structure for the Mn sites was generated from the experimentally determined configuration of the prototype (Fig. S14), and it was compared with a relaxation where a ferromagnetically ordered state had been imposed. The relaxed AFM spin structure is consistent with the experimentally observed CE-type magnetic ordering, but the additional incommensurate modulation cannot be modelled within the periodic boundary conditions of DFT."

Figure S14 however shows noncollinear magnetic structures. Does the non-collinearity not affect the energy state?

3. I raised another question why the magnetic field effect was not investigated. They say, "our 134 prototype system may not exhibit CMR as it is unlikely to show the perquisite phase coexistence known to be intrinsic to this phenomenon."

Unfortunately, I do not understand what the authors would like to say. Anyhow, I am still skeptical if the CMR effect in the perovskite Mn oxide compounds with no Mn ions on A sites can be discussed based on the present study.

4. The authors add the temperature dependence of the monoclinic angle in Fig. S8. I am rather skeptical that the angle is determined with an error smaller than 0.001 degrees, as shown in the top right and bottom left panels.

5. The peak (4 4 -2) in the top right panel of Fig. 1(b) may not be correctly indexed.

Reviewer #2 (Remarks to the Author):

The authors have adequately addressed previous comments. The paper can now be recommended for publication.

We thank the Reviewer #1 for a further set of detailed comments. We have greatly improved how we present the magnetic structures and associated diffraction data, as well as clarifying how we construct the co-linear model used for the DFT calculations. Detailed replies are given below.

Reviewer #1 (Remarks to the Author):

1. I gave a comment on the magnetic structure of the (Ca,Na) solid solution in my previous report. The authors respond that the system has two magnetic structures: the famous CE-type and an additional incommensurate magnetic structure. They revise a section as follows:
"Our neutron diffraction data (Fig. \ref{structure}(a) and Fig. S14), shows that the well know CE-type antiferromagnetic order, that is intrinsically coupled to the orbital ordered state, is also observed in our half-doped prototype system ($A = \text{Na}$). There is a steady decrease in this order (see Fig. S14) with the frustration of the magnetic interactions resolving itself in an incommensurate modulation towards $x = 3/8$."

I still do not understand how the neutron diffraction data were analyzed to pin down the magnetic structures. The neutron diffraction data are provided as heat maps both in the Figs. 2(a) and S14. I am afraid that such a heatmap is not suitable for understanding the magnetic structure. For example, 'a steady decrease in this order' is not clearly shown. I recommend that the raw diffraction patterns are shown in the conventional way with each peak indexed.

⇒ We have now included additional Rietveld fit of the commensurate magnetic structure as part of Fig S14 for $A = \text{Na}_{0.9}\text{Ca}_{0.1}$ and $A = \text{Na}_{0.4}\text{Ca}_{0.6}$. We clearly show the peak indexing of the magnetic Bragg peaks. The steady decrease of this order is shown in panel (a) of the figure where the amplitudes of the corresponding modes (extracted from the Rietveld plots that are now shown) is summarised.

In addition, the mS2 and mM2+ modes in Fig. S14 are not defined, which would be confusing.

⇒ We add additional panels to Fig. S14 to show the mS2 and mM2+ modes. We also include the full order parameter directions (as defined in ISODISTORT) associated with the magnetic modes.

The schematics in the inset are also confusing due partly to the superposition of two antiferromagnetically coupled 'layers'. It is important because the incommensurate pseudo-CE-type structure proposed by Johnson et al. has ferromagnetic stacking for some sites while antiferromagnetic stacking for the other sites.

⇒ Our figures show AFM interaction in the stacking direction (as evident for the overlap of head and tail of the arrows.) We clarify this point in the caption " The commensurate part of the magnetic structure in the **bc** plane is visualized in the inset, where the interactions along the out-of-plane **a** direction are AFM." We also included additional projections of the magnetic structures in a new panel (d). We only plot/consider the commensurate part of the magnetic structure in the present study and hence the difference between that presented by Johnson et al. While the nature of the incommensurate magnetic ordering is

interesting, it cannot itself be the origin of the commensurate orbital order that we discuss here and so discussing it further would be a distraction to the main thrust of our manuscript.

In the AMn7O12 system, the Mn moments on A sites can also contribute to the magnetic order. In the inset, however, Mn moments on A sites is, perhaps, omitted.

- ⇒ As stated in the caption the A-site ordering occurs at the X point. We do not plot the A-site magnetic moments in the present figures. However, we now included a visualisation of the magnetic mode (mX3+) associated with the A-sites magnetic ordering. Note that this ordering is modulated by the 1(non-magnetic):3(magnetic) A-site cation ordering mode M1+ as described in the main paper.

We have also already included the o Neutron data for all sample in the Figshare repository for anyone wishing to reproduce these results

I do not understand what red lines attached to some Mn ions in the inset denote, either. The correspondence between the magnetic structure and orbital order is not clear, either.

- ⇒ The red lines indicated the JT long axes (as they do in the main paper). This point is clarified in the caption now. With this information the correspondence between magnetic structure and orbital order can be appreciated.

If the authors suggest the coexistence of two or more magnetic phases in some compounds in some temperature ranges, multiple orbital states may also coexist.

- ⇒ We do not suggest the coexistence of two or more magnetic phases, we state that there is an incommensurate component that grows in amplitude towards the 3/8th doping level. Unlike Johnson et al., we do not find any evidence that there is any phase coexistence. It is likely in his study that the phase coexistence arises from inhomogeneity during sample preparation at high pressure. The magnetic Bragg peaks observed in our study all belong to the same phase for a given composition. We stress the high crystallinity and single-phase nature of our sample at several point throughout the manuscript and demonstrate this in Table S1-1, Fig S9 and S10.

Finally, the draw data from which the magnetic structure are derived is already provided in the electronic repository (DOI: 10.6084/m9.figshare.14823678) for anyone wish to repeat our analysis.

2. I raised a question about the consistency between the neutron data and DFT+U calculation. The authors reply that the commensurate part was reproduced by DFT+U while the incommensurate part was not. I understand that the boundary condition for the calculation is harmful in the incommensurate case. They revise a section as follows:

"A collinear antiferromagnetically ordered spin structure for the Mn sites was generated from the experimentally determined configuration of the prototype (Fig. S14), and it was compared with a relaxation where a ferromagnetically ordered state had been imposed. The relaxed AFM spin structure is consistent with the experimentally observed CE-type magnetic ordering, but the

additional incommensurate modulation cannot be modelled within the periodic boundary conditions of DFT."

Figure S14 however shows noncollinear magnetic structures. Does the non-collinearity not affect the energy state?

The reviewer is quite right to point this out. It is in fact probable that if we went to a higher level of theory in our calculations (including spin orbit coupling) we could achieve an even greater level of agreement between the DFT relaxed structure and our experimental results. However, the current level of agreement considering only spin polarisation, where we approximate with a collinear structure that captures the dominant exchange interaction, appears to be very good already

We make the following changes to the manuscript to highlight this:

"Qualitatively similar results are obtained for a wide range of U and J. The reduced mode ratio means that the electron density in the OO layers bleeds out slightly more into the CD than implied experimentally. **The origins of these small quantitative disagreements could be due to the fact we consider only one average U and J for all Mn sites, and that we do not model the experimentally observed spin canting and additional incommensurability within the present level of theory.** "

and

"A collinear antiferromagnetically ordered spin structure for the Mn sites was generated from the experimentally determined configuration (Supplementary Fig. 14), **with the smaller canting along $\langle \text{textbf{c}} \rangle$ ignored in this approximation.**"

3. I raised another question why the magnetic field effect was not investigated. They say, "our 134 prototype system may not exhibit CMR as it is unlikely to show the perquisite phase coexistence known to be intrinsic to this phenomenon."

Unfortunately, I do not understand what the authors would like to say. nyhow, I am still skeptical if the CMR effect in the perovskite Mn oxide compounds with no Mn ions on A sites can be discussed based on the present study.

⇒ CMR in the manganites is understood to occur due to intrinsic segregation between insulating AFM OO states and conduction FM phases. This phase segregation is evident as a function of temperature and applied magnetic field in the narrow band manganite LCMO and LPCMO discussed in the introduction. As laboured in our manuscript we never observe any phase segregation in our prototype systems, and if he had, the detailed structural work identifying a novel OO state that we have done would not have been possible!

We demonstrate that at the zero and $\frac{1}{2}$ doping level in our prototype system that the OO/CO phases are consistent with those observed in the canonical systems. It is hence extremely likely that the novel OO with CD state we observe at the $\frac{3}{8}$ doping level is of upmost relevance to CMR in LCMO /LPCMO systems that occurs precisely at this level. Furthermore, we show that our model is consistent with experimentally observed

superstructure peaks in LPCMO, and that it is stable to relaxation under DFT+U in a doped LaMnO₃ system with the key features of the electronic structure (Fig S15) essentially very similar to our prototype system.

We have thus firmly established a link between our prototype and the insulating state of the canonical system.

We clarify this point in the concluding remarks now:

“In conclusion, we have shown that $\text{AMn}_{3-x}\text{Mn}_4\text{B}\text{O}_{12}$ $A = \text{Na}_{1-x}\text{Ca}_x$ and $\text{La}_{1-x}\text{Ca}_x$ can act as a prototype system for canonical CMR perovskites, having the same electronic orderings observed in LaMnO₃ at zero and half-doped levels, and through the correspondences observed in, DFT relaxations and their diffraction data. Detailed crystallographic investigations of this prototype system has allowed us to identify a new kind of orbital order at the $x = \frac{3}{8}$ doped level of the manganite phase diagram consisting of OO and CD stripes.”

4. The authors add the temperature dependence of the monoclinic angle in Fig. S8. I am rather skeptical that the angle is determined with an error smaller than 0.001 degrees, as shown in the top right and bottom left panels.

⇒ The reviewer is quite right to point this out. In fact, we did not originally include the beta angles since, on account of the pseudo-symmetry in $A=\text{Na}_{0.4}\text{Ca}_{0.6}$ and $\text{Na}_{0.5}\text{Ca}_{0.5}$, making them metrically tetragonal, these values refine to 90 within the resolution of the experiment / sample. However, on request of referee 1 and 2 these have subsequently been added in. The plotted small changes in beta (6 thousands of degree) as a function of temperature are essentially noise. A better understanding of the degree of pseudo symmetry in these samples can be gained by studying Fig S9 where we monitor the change in refined R_{wp} as a function of strain transforming as Γ_5^+ (corresponding with the beta angle). This figure shows that the macro strain associated with the sheer type distortion is $\sim 0.02\%$ which is essentially the same as the microstrain (which in itself is exceptionally low for these samples). In the paper we hence establish with a high degree of precision the pseudo symmetry in the $A=\text{Na}_{0.4}\text{Ca}_{0.6}$ We emend caption of Fig S8 to further emphasise these point.

“We note that for $A=\text{Na}_{0.4}\text{Ca}_{0.6}$ and $\text{Na}_{0.5}\text{Ca}_{0.5}$, the refined variation in β which is of the order of 6 thousands of a degree across the temperature range, is likely just experimental noise. A better appreciation of the intrinsic resolution of the current experiment can be appreciated from Fig S9 that established the sheer monoclinic (Γ_5^+) microstrain as being no larger than the microstrain in the samples, and hence we refer to these phase as metrically pseudo-tetragonal.

5. The peak (4 4 -2) in the top right panel of Fig. 1(b) may not be correctly indexed.

⇒ Many thanks for spotting this. It should be (4 2 -2). This is now change.

REVIEWERS' COMMENTS

Reviewer #1 (Remarks to the Author):

I have found that the authors address most of the comments that I raised in the second round.

To my comment 3, they respond as

"We demonstrate that at the zero and $1/2$ doping level in our prototype system that the OO/CO phases are consistent with those observed in the canonical systems. It is hence extremely likely that the novel OO with CD state we observe at the $3/8$ doping level is of utmost relevance to CMR in LCMO /LPCMO systems that occurs precisely at this level."

Although I do not fully agree with their above-mentioned opinion, the clear claim itself is in my opinion worth publishing in Nature Communications. It may be a good idea to leave the validity of their conclusion to the judgement of the audience.

Response to Referee #1:

Referee #1 made the final comments on our revised manuscript:

“

I have found that the authors address most of the comments that I raised in the second round.

To my comment 3, they respond as

"We demonstrate that at the zero and 1/2 doping level in our prototype system that the OO/CO phases are consistent with those observed in the canonical systems. It is hence extremely likely that the novel OO with CD state we observe at the 3/8 doping level is of upmost relevance to CMR in LCMO /LPCMO systems that occurs precisely at this level."

Although I do not fully agree with their above-mentioned opinion, the clear claim itself is in my opinion worth publishing in Nature Communications. It may be a good idea to leave the validity of their conclusion to the judgement of the audience.

“

We thank the referee for acknowledging the value of our contribution, which among other things identifies a new prototype system that can be used for gaining insight into the orbital ordered states in the CMR manganites. We are sure our work will generate a healthy amount of discussion within the literature, and we hope that others will explore the extend to which these 134 perovskites can be used as prototype systems for studying charge and orbital ordering phenomena.